# Molecular Pathogenesis of Endotheliopathy and Endotheliopathic Syndromes, Leading to Inflammation and Microthrombosis, and Various Hemostatic Clinical Phenotypes Based on “Two-Activation Theory of the Endothelium” and “Two-Path Unifying Theory” of Hemostasis

**DOI:** 10.3390/medicina58091311

**Published:** 2022-09-19

**Authors:** Jae C. Chang

**Affiliations:** Department of Medicine, University of California Irvine School of Medicine, Irvine, CA 92868, USA; jaec@uci.edu; Tel.: +949-943-9988

**Keywords:** combined micro–macrothrombosis, endotheliopathy, hemostasis, inflammation, microthrombosis, macrothrombosis, thrombocytopenia, thrombosis, vascular microthrombotic disease

## Abstract

Endotheliopathy, according to the “two-activation theory of the endothelium”, can be triggered by the activated complement system in critical illnesses, such as sepsis and polytrauma, leading to two distinctly different molecular dysfunctions: (1) the activation of the inflammatory pathway due to the release of inflammatory cytokines, such as interleukin 6 and tumor necrosis factor-α, and (2) the activation of the microthrombotic pathway due to the exocytosis of hemostatic factors, such as ultra-large von Willebrand factor (ULVWF) multimers and FVIII. The former promotes inflammation, including inflammatory organ syndrome (e.g., myocarditis and encephalitis) and multisystem inflammatory syndrome (e.g., cytokine storm), and the latter provokes endotheliopathy-associated vascular microthrombotic disease (VMTD), orchestrating thrombotic thrombocytopenic purpura (TTP)-like syndrome in arterial endotheliopathy, and immune thrombocytopenic purpura (ITP)-like syndrome in venous endotheliopathy, as well as multiorgan dysfunction syndrome (MODS). Because the endothelium is widely distributed in the entire vascular system, the phenotype manifestations of endotheliopathy are variable depending on the extent and location of the endothelial injury, the cause of the underlying pathology, as well as the genetic factor of the individual. To date, because the terms of many human diseases have been defined based on pathological changes in the organ and/or physiological dysfunction, endotheliopathy has not been denoted as a disease entity. In addition to inflammation, endotheliopathy is characterized by the increased activity of FVIII, overexpressed ULVWF/VWF antigen, and insufficient ADAMTS13 activity, which activates the ULVWF path of hemostasis, leading to consumptive thrombocytopenia and microthrombosis. Endothelial molecular pathogenesis produces the complex syndromes of inflammation, VMTD, and autoimmunity, provoking various endotheliopathic syndromes. The novel conceptual discovery of in vivo hemostasis has opened the door to the understanding of the pathogeneses of many endotheliopathy-associated human diseases. Reviewed are the hemostatic mechanisms, pathogenesis, and diagnostic criteria of endotheliopathy, and identified are some of the endotheliopathic syndromes that are encountered in clinical medicine.

## 1. Introduction

Endotheliopathy is a very common vascular disorder that develops due to functional and/or anatomical changes in the endothelium, which results in serious molecular responses that produce a variety of human diseases. This condition is triggered by the activated complement system secondary to external insults, such as the toxin of pathogen and polytrauma, or to internal physiologic changes, such as pregnancy and autoimmune disease [1,2,3]. It promotes two endothelial molecular pathways: inflammatory and microthrombotic [4,5,6,7,8]. Both inflammation and microthrombosis occur commonly in sepsis and critical illnesses [4,5,6,7,8,9,10] due to the released molecules from endotheliopathy caused by the toxin of pathogen, drug, poison, chemical, vaccine, or venom, and from functional changes after surgery, transplant, or polytrauma [5,6]. Inflammation and disseminated vascular microthrombotic disease (VMTD) often simultaneously occur, and they cause a variety of phenotypes of clinical pathology. The endothelial damage provokes major molecular and biological events, and it produces a spectrum of clinical disorders, from focal, multifocal, localized, and disseminated disease to multisystem inflammatory syndrome (MOIS) and multiorgan dysfunction syndrome (MODS), not only due to the cellular response and hemostatic activation, but also due to the delayed autoimmune response [11,12,13,14].

Until recently, both the pathophysiological mechanisms of inflammation and microthrombosis had not been clearly understood. Because of the concurrent occurrence of inflammation and thrombosis [15,16], the hypothesis of immunothrombosis, which supports the cross-talk between inflammation and thrombosis, was introduced as a newly proposed hemostatic mechanism to expand the concept of macrothrombosis based on the molecular event of neutrophil extracellular traps (NETs) [17,18,19]. Immunothrombosis was thought to be the host defense mechanism to limit the systemic spreading of the pathogen in sepsis [18]. However, despite extensive research efforts to find an interaction between inflammation and thrombosis, no persuasive evidence supporting the “cross-talk theory” has been materialized. Furthermore, during the current COVID-19 pandemic, the concept of the previously defined disseminated intravascular coagulation (“DIC”), which was thought to play a critical role in coagulation dysfunction, was proven to be incorrect because disseminated intravascular “microthrombosis” is confirmed to be the true pathology of sepsis-associated coagulopathy [6,8,10,20].

According to recently proposed “two-activation theory of the endothelium”, depicted in Figure 1, endotheliopathy leads to the activation of two molecular pathways, “inflammatory” and “microthrombotic”, promoting inflammation and microthrombosis, respectively [4,5,6]. This theory delineates that two pathways are completely independent, but are concurrent events that promote two different molecular pathogeneses. Additionally, the “two-path unifying theory” of hemostasis in vivo (Figure 2), the genesis of microthrombosis (i.e., VMTD), was established to be the result of partial hemostasis triggered by ultra-large von Willebrand factor (ULVWF) path of hemostasis. In disseminated endotheliopathy, inflammation and microthrombogenesis often coexist, but they each promote separate pathogeneses, producing many distinctly different and combined clinical phenotypes [6,10].

This author will concisely review how endotheliopathy is initiated, and also what causes the molecular pathogenesis, based on two mutually congruous hemostatic theories in the vascular wall model: (1) the “two-activation theory of the endothelium”, illustrating the genesis of inflammation and microthrombosis (i.e., VMTD) [4], and (2) the “two-path unifying theory” of hemostasis, identifying the mechanisms of microthrombosis and macrothrombosis (e.g., deep venous thrombosis (DVT)) [11,12]. Because disseminated microthrombosis produces much more serious clinical disorders that determine the outcome of endotheliopathy than inflammation, this article will largely be focused on the pathogenesis and clinical feature of EA-VMTD.

## 2. Genesis of Endotheliopathy

The endothelium is a unique single layer of cells lining between the blood in circulation and the subendothelial tissue of every organ to protect the body from the intrusion of harmful external toxic molecules and microorganisms. It also provides the direct and indirect contact of nutrients and metabolic components to the subendothelial and extravascular tissues to sustain life, and it is the barrier to and partner with the blood cells to maintain hemostasis and physiologic homeostasis. Therefore, an insult causing endothelial damage disrupts the normal anatomy and physiologic function. The endothelium may lead to diversified paths of the pathogenesis that trigger the dysfunction of the cells, tissues, organs, and multisystem. Until now, clinical medicine has designated most human diseases with terms that define and appropriate the dysfunction of the cell (e.g., microangiopathic hemolytic anemia), tissue (e.g., rhabdomyolysis), organ (e.g., acute kidney failure), and multisystem (e.g., systemic lupus erythematosus). However, clinical medicine has not been able to put together the etiologic identities and pathogeneses of many clinical syndromes in relation to molecular, metabolic, and structural alteration. This paucity of our knowledge in endotheliopathy has also entailed the utilization of imprecise terms for diagnosis, such as Kawasaki disease, acute respiratory distress syndrome (ARDS), multisystem inflammatory syndrome in child (MIS-C), thromboangiitis obliterans (e.g., Buerger’s disease), cerebral venous sinus thrombosis (CVST), DIC-like coagulopathy, limb gangrene, and others, to define the clinical syndromes that have been occurring during the COVID-19 pandemic and in other diseases [10].

### 2.1. Causes of Endotheliopathy

Endotheliopathy is not a distinctive disease that delineates the dysfunction of a specific cell type, tissue, organ, or system, but an intermediary pathology that leads to almost every human disease involving the vascular-tree system. To date, the accepted medical tenet has been that endothelial damage results in a transient endothelial intravascular injury, typically due to localized vascular pathology, and often with limited clinical consequence. However, now it is abundantly clear that “endotheliopathy” could become a major pathology that produces life-threatening cellular, tissue, and organ damage, leading to their dysfunction, and even to multisystem disease [4,5,6,7,8,9,10]. The endothelium is the most important structure protecting lives, in which the pathology of many human diseases could originate or be influenced. Its integrity often determines the clinical outcomes of critical illnesses.

#### 2.1.1. Complement System

Endotheliopathy is triggered by the activated complement system in response to pathogens and other insults [3,6,21]. Complement activation can occur through one of three different pathways (i.e., classical, alternate, and lectin). The complement system can not only be activated by an external insult, such as a pathogen, toxin or drug, vaccine, venom, surgery, transplant, or polytrauma, but also by internal physiologic changes, such as pregnancy, hypertension, hyperglycemia, as well as the autoimmune process [5,6,22,23].

The lytic function of the complement membrane attack complex (MAC: C5b-9) serves as an important defense against bacteria, viruses, and parasites, and the genetic deficiency of the terminal complement proteins C5 through C9 predisposes the host to recurrent infections [24]. The innate immune response of complement activation is not only the first step in protecting the host against a variety of pathogens, but it may also play a detrimental role in innocent bystander endothelial cells (ECs). Should the unchecked production of MAC be deposited to the endothelial membrane, channel formation could be formed and lead to endothelial dysfunction in the host [3]. When MAC provokes either structural or functional endothelial damage in the host, it activates at least two unique molecular mechanisms, which promote clinically significant endotheliopathic syndromes [4,5,6,7].

#### 2.1.2. Endothelial Molecular Pathogenesis

The molecular pathogenesis of endotheliopathy is elaborated in the “two-activation theory of the endothelium”, which shows complement-induced endothelial molecular events (Figure 1), leading to the activation of: (1) the inflammatory pathway, and (2) the microthrombotic pathway [4]. MAC-induced endotheliopathy is suspected to occur if the endothelium is “unprotected” due to downregulated CD59 [25,26].

Both functional and structural injuries of the endothelium release several mediators altering homeostasis and activate a host of biomolecules expressed in ECs, which can promote, directly and indirectly, the creation of different pathologic and clinical phenotypes. The numerous biomolecules that are released from ECs include inflammatory cytokines, hemostatic factors, and adhesive molecules, which are interleukins (IL), tumor necrosis factors (TNF), interferons (IFN), coagulation factors, tissue plasminogen activator, prostacyclin, nitric oxide, and others [6,7,27]. Although their roles in vascular disease have not been well defined yet, the pathogenesis of inflammation is postulated to be due to released cytokines and chemokines, such as interleukins IL-1, IL-2, IL-6, TNF, IFN, and others, and microthrombogenesis is promoted by ULVWF multimers and hemostatic factors, resulting from endotheliopathy [5,6,7,8,9,10,28]. These molecular pathogeneses were predicted before the onset of the COVID-19 pandemic [6,7], and important molecular events have been unequivocally confirmed during the pandemic [29,30].

The mechanism of molecular pathogenesis in disseminated arterial and venous endotheliopathy based on the “two-activation theory of the endothelium” is shown in Figure 1. On the one hand, the activated inflammatory pathway provokes inflammation due to released inflammatory cytokines in sepsis, or other critical illnesses. On the other hand, the activated microthrombotic pathway triggers the excessive release of ULVWF/FVIII from endothelial exocytosis [5,6]. Should a relative insufficiency of ADAMTS13 be present due to the heterozygous gene mutation or polymorphism of the gene, and/or due to the excess release of ULVWF multimers, partial hemostasis of lone ULVWF path becomes activated and ULVWF attract platelets, which leads to the formation of “microthrombi strings” composed of platelet–ULVWF complexes [5,6]. This process is called microthrombogenesis, and leads to endotheliopathy-associated disseminated vascular microthrombotic disease (EA-VMTD) [5,6,11].

This partial hemostasis due to lone activation of the ULVWF path is characterized by consumptive thrombocytopenia, overexpressed ULVWF/VWF/VWF antigen and increased activity of FVIII due to their endothelial release, resulting in relative ADAMTS13 insufficiency. Endotheliopathy is systemic in sepsis and disease due to certain drugs or toxins, and it is clinically disseminated in the microvasculature, promoting “microvascular microthrombosis” (i.e., EA-VMTD). In the past, the “coagulopathy” occurring in the septic patient was thought to be due to the overactivation of the tissue factor (TF) path leading to systemic “fibrin clots” formation, and had been called “DIC”. However, its pathogenesis has never been clearly established. Based on the two novel hemostatic theories, “DIC” has been identified as a form of thrombosis, which is exclusively due to “microthrombosis” caused by the activated ULVWF path that develops from the injury of ECs [5,6,8,20]. The conflict in the terms between “coagulation” and “thrombosis” related to “hemostasis” has contributed to the misunderstanding of the pathogenesis of “DIC”, and of the identification of the pathophysiological mechanism of hemostasis in vivo. Thus, in the past, the true meaning and genesis of “fibrin clots” and “microthrombi” could not been conceptualized, which contributed to uncomfortable debates between coagulopathy and thrombosis (thrombopathy) in the coagulation community. To clarify the different meanings of their conceptual terms, I have briefly summarized the discriminating interpretations of the terms in Table 1.

Many medical studies have emphasized that the increased activities of FVIII and VWF in critical illnesses are reactive changes in response to inflammation promoting the “hypercoagulable state” of “DIC”. This interpretation has led to a conceptually and structurally wrong conclusion on the nature of microthrombi because the increased expressions of FVIII and VWF are always the result of endothelial exocytosis from endotheliopathy, which is perfectly consistent with vascular injury limited to ECs producing “microthrombi strings”, as shown in the “two-path unifying theory” of hemostasis (Figure 2) [11,12].

### 2.2. Genesis of Clinical Phenotypes in Endotheliopathy

Endotheliopathy, even generalized, only activates lone ULVWF path of hemostasis because the complement-activated endothelial damage that releases biomolecules due to the pathogen toxin is limited to ECs, without affecting the subendothelial tissue (SET) and extravascular tissue (EVT) [6]. Disseminated microthrombosis is made of microthrombi strings in the microvasculature because SET/EVT damage that releases TF to the intravascular space does not occur, and fibrin clots are not produced to form complete blood clots (i.e., macrothrombosis), as presented in Figure 2.

In contrast, a bleeding intravascular injury that causes both local EC and SET/EVT damage typically occurs in local trauma following local vascular damage. An in-hospital vascular access/device, or indwelling central vascular catheter, which releases not only a small amount of ULVWF/FVIII from local ECs, but also releases sufficient TF from SET and EVT, produces macrothrombosis. The relationship between ULVWF release from ECs and TF release from ECs and SET/EVT from their respective damage is illustrated in Figure 3. In sepsis, endotheliopathy is disseminated even though its extent and phenotype expression are dependent upon endothelial heterogeneity and the tropism of the pathogen. When microthrombogenesis is triggered, disseminated microthrombosis causes clinical phenotypes by affecting different organs and producing organ dysfunction syndrome, including MODS. Furthermore, more complex phenotypes of microthrombosis can occur depending on: (1) the extent of the vascular-tree involvement (e.g., the multiplicity of the tissues and organs), (2) the localization of different vascular functional characters (e.g., arterial vs. venous system), and (3) an additional interacting mechanism with an underlying gene mutation or environmental factor (e.g., thrombophilia, such as FV Leiden, or protein C deficiency, and von Willebrand disease, pathogens, and toxins) [13]. All of these different phenotypes that occur in endotheliopathy-associated “focal”, “multifocal”, “localized”, “regionalized”, and “disseminated” microthrombosis are inclusively called EA-VMTD [5].

#### 2.2.1. Hemostasis Based on the Blood-Vessel Model

The concept of EA-VMTD cannot be understood without the comprehension of the hemostatic mechanism in vivo. However, hemostasis based on the blood-vessel model is simple and logical if the anatomy and physiology of the blood-vessel wall are understood, as illustrated in Figure 3, and the three fundamentals in hemostasis are applied to the thrombogenesis, as summarized in Table 2. The first and second principles illustrate how the molecular event of hemostatic components is initiated in the vascular-wall injury, and the third principle elaborates the physiologic paths of hemostasis that produces the pathologic phenotypes following an intravascular injury [7].

#### 2.2.2. Vascular Injury Provoking Thrombosis

The established dogma of Virchow’s triad on thrombosis and thrombogenesis, which includes: (1) endothelial injury, (2) the stasis of blood flow, and (3) hypercoagulability, is considered the essential three components initiating the formation of a thrombus [31]. Unfortunately, this conceptual misunderstanding of hemostasis, thrombosis, and coagulation has not only confused the concepts and uses of the terms, “coagulation” and “thrombosis”, but it has also hampered the discovery process of the true hemostatic mechanism in vivo to date. This author has attempted to redefine the conceptual connotation of these terms, as shown in Table 1.

After the reinterpretation of “DIC” utilizing the differentiating concepts between “coagulation” and “thrombosis”, and their implied meanings in relation to hemostasis in vivo and coagulation in vitro, the true identity of “DIC” was found to be “endotheliopathic microthrombosis” rather than the “uncontrolled” activation of coagulation in vitro or in vivo [8,20]. In other words, “DIC” is an endothelial phenomenon caused by well-defined partial hemostatic process (i.e., ULVWF path) rather than vague in vivo coagulation disorder reflecting in vitro coagulation abnormalities. Thus, “DIC” has been newly defined as EA-VMTD, which explains not only microvascular microthrombosis (i.e., VMTD), but also confirms the molecular changes associated with the disease (i.e., thrombocytopenia, increased activity of FVIII, and overexpression of ULVWF/VWF antigen) [5,6,8,20]. Furthermore, this author was able to link EA-VMTD (microthrombosis of endotheliopathy) to “DIC” (microthrombosis of sepsis) and could separate the “fibrin clots” of fibrin clot disease (i.e., acute promyelocytic leukemia) from the “microthrombi strings” of “DIC”, constructing two hemostatic paths in vivo based on the vascular-wall model after analyzing the following inexplicable hemostatic facts. Finally, identified are the true characters of “microthrombosis” and “fibrin clots” according to the “two-path unifying theory” of hemostasis [11,12].

Microthrombi strings composed of platelet–ULVWF complexes [32,33];Documented endothelial exocytosis of ULVWF and FVIII in patient with sepsis and other endotheliopathy [34,35,36];The well-known role of ULVWF from ECs in the early phase of hemostasis in external bodily injury;Prominent interaction between platelets and ULVWF in a vascular injury, resulting in consumptive thrombocytopenia [37,38];The lack of participation of TF in hemostasis producing septic microthrombosis [4];TF unavailable in the EC injury but available from the SET/EVT in local bleeding vascular injury;Distinctly different disseminated microthrombosis occurring in the microvasculature in sepsis compared with localized macrothrombosis formed after fibrin clots occurring in a local large-vessel injury;The irrefutable hemostatic fundamental: “The hemostasis and thrombogenesis can be activated only by a vascular injury”.

The three fundamentals in normal hemostasis are perfectly congruous with the “two-path unifying theory” [7]. Further, the above factual components demonstrate that it is impossible to correctly redefine both “microthrombosis” and “macrothrombosis” using the contemporary hemostatic theory which is based on sequential activation via the extrinsic coagulation cascade initiated by the TF–FVIIa complex alone, [11], as well as fibrin clot disease of acute promyelocytic leukemia (APL) [39]. The “two-path unifying theory” of hemostasis can explain every hemostatic phenotype that occurs in microthrombosis, macrothrombosis, and the unique fibrin clot disease. A true in vivo hemostatic theory should be inclusive of the following molecular and physiologic changes of thrombogenesis [6,10,13,14]:Damaged ECs from the blood-vessel wall following a vascular injury release ULVWF/FVIII in partnership, which are the essential components, along with platelets, in the formation of microthrombi strings. The process of ULVWF interacting with platelets is called “microthrombogenesis”;Damaged SET from the blood-vessel wall following a local vascular injury releases TF into circulation, which activates FVIIa and leads to the formation of fibrin clots via the extrinsic cascade with the sequential activation of FIX, FX, FV, FII, and fibrinogen to fibrin. The process is called “fibrinogenesis”;A local traumatic vascular injury involving ECs and SET produces both microthrombi strings and fibrin clots. The proposed theory is that the forming process of macrothrombosis must be the result of the “unifying mechanism of microthrombi strings and fibrin meshes”. This process is called “macrothrombogenesis”, in which neutrophil extracellular traps (NETosis) passively participate;The understanding of this unifying mechanism is very important in the understanding of arterial and venous combined micro–macrothrombosis, which is a process that should be called micro–macrothrombogenesis.

The above three basic physiological mechanisms of hemostasis in vivo, constructed in Figure 2 and Table 3, can not only explain every hemorrhagic disease and thrombotic disorder, but it can help also interpreting additional complex hemostatic phenotypes in stroke, heart attack, venous thromboembolism (VTE), CVST, gangrene syndromes, and other tissue/organ-specific syndromes [10,40]. Because EC damage is disseminated without SET damage in sepsis, but EC and SET damages are deeper but limited to localized vascular trauma, the microthrombi strings causing microthrombosis must be the result of microthrombogenesis triggered by endotheliopathy, but macrothrombus unified of microthrombi strings and fibrin clots is the result of fibrinogenesis and macrothrombogenesis in locality (Table 3). Two hemostatic theories have been identified: (1) the “two-activation theory of the endothelium” to explain inflammation and microthrombosis (e.g., VMTD), and (2) the “two-path unifying theory” of hemostasis to explain the geneses of microthrombosis, fibrin clot disease, and macrothrombosis (e.g., DVT). The former is illustrated in Figure 1, and the latter in Figure 2. These two hemostatic theories are being applied in identifying the activated hemostatic mechanisms in the different phenotypes of hemostatic diseases. These mechanisms have correctly established the pathogeneses of many poorly defined thrombotic diseases, which include microthrombosis (e.g., septic endotheliopathy, ARDS, HUS, veno-occlusive disease (VOD) [4,5,6,7,8,9,10], macrothrombosis (e.g., acute ischemic stroke, deep venous thrombosis) [10,12,40], arterial combined micro–macrothrombosis (e.g., symmetrical peripheral gangrene) [6,10], venous combined micro–macrothrombosis (e.g., VTE, CVST) [13,14], MODS [6], and ARDS, with complex forms of thrombosis during the COVID-19 pandemic [7,10], including the following endotheliopathic syndromes in other human diseases.

## 3. Endotheliopathic Syndromes

Endotheliopathy produces an array of clinical and pathologic phenotypes among the endotheliopathic syndromes, which include inflammation, vascular microthrombotic diseases, autoimmune diseases, and combined inflammatory and microthrombotic syndromes, as illustrated in the projected distribution of the endotheliopathic syndromes in Figure 4. In the figure, the geneses of several endotheliopathic phenotypes are known or suspected to be the result of the intersecting relationship between the two molecular pathogenetic mechanisms of inflammation and microthrombogenesis. An emphasis will be placed on the endotheliopathic syndromes orchestrated by the pathogenetic mechanism of EA-VMTD to assist the clinician in identifying inexplicable hemostatic diseases and their pathogeneses.

### 3.1. Consumptive Thrombocytopenia

Isolated thrombocytopenia occurs not uncommonly in critically ill patients [41,42], and it has often been termed ITP (i.e., acute, chronic, idiopathic, or immune thrombocytopenia), or thrombocytopenia in critically ill patients (TCIP) [4]. In the past, and during the COVID-19 pandemic, most cases of the thrombocytopenia of unknown origin were considered ITP (immune), including TCIP, gestational thrombocytopenia [43], and ITP-like syndrome in vaccine side effects [13,14], depending on the clinical phenotypes and underlying pathologies, as shown in Figure 5. Surprisingly, TTP-like syndrome, which is characterized by consumptive thrombocytopenia, also occurs in the same critically ill patients associated with additional hematologic phenotypes of microangiopathic hemolytic anemia (MAHA) and MODS [5]. This author has proposed that most of the ITP and TTP-like syndromes are caused by endotheliopathy, which releases ULVWF multimers that could activate ULVWF path of hemostasis, leading to microthrombogenesis, and produces microthrombi strings composed of platelet–ULVWF complexes [14]. Thrombocytopenia is the result of platelet consumption in both ITP- and TTP-like syndromes. The difference is that ITP occurs in venous endotheliopathy, but TTP-like syndrome develops in arterial endotheliopathy, as summarized in Table 4 [13,14].

Therefore, some ITP should have been called “ITP-like syndrome”, which is endotheliopathy-associated consumptive thrombocytopenia different from immune destructive thrombocytopenia caused by platelet antibodies. Both ITP-like syndrome and TTP-like syndrome are the same disease due to endotheliopathy, but occurring in different milieux. ITP-like syndrome occurs predominantly in the venous system, and TTP-like syndrome occurs in the arterial system. This concept has a profound implication in the understanding of the immune mechanism and endothelial mechanism on the pathogenesis of thrombocytopenia in clinical medicine as shown in Figure 5.

In the recent past, the vascular diseases affecting the endothelium have included critical illnesses, atherosclerosis, vasculopathy, microvascular microthrombosis, vascular thrombosis, vasculitis, angiitis, endotheliitis, microangiopathy, angiomatosis, telangiectasia, and other vessel-related congenital and acquired diseases. Not surprisingly, some of them have been associated with inexplicable thrombocytopenia. Since the “two-path unifying theory” of hemostasis is proposed, disseminated EC damage with/without extending into SET has been suspected to be a promoting factor of consumptive thrombocytopenia, as shown in Figure 5. This theory supports that thrombocytopenia is an early sign of endotheliopathy in disseminated or regional vascular disease, and the evidence of the activated ULVWF path of hemostasis can be confirmed easily by endothelial markers, such as the overly expressed ULVWF/VWF antigen and the elevated activity of FVIII, and especially with insufficient ADAMTS13 activity [5,6]. Consumptive thrombocytopenia is ubiquitous when the ULVWF path of hemostasis is activated in critical illnesses based on microthrombogenesis shown in Figure 1. Additionally, two clearly distinguishable types of endotheliopathy, arterial and venous, have been identified, as shown in Table 4, which explains the variable clinical manifestations of thrombocytopenia that occur in consumptive thrombocytopenia [13,14].

### 3.2. Arterial Endotheliopathy vs. Venous Endotheliopathy

The circulatory system can be divided into the arterial system and venous system. Each is characterized by differences in anatomy, hemodynamics, physiology, and functionality. Their differences have important implications in the formation of clinical phenotypes in localized and disseminated thrombosis between the arterial and venous systems. Both arterial endotheliopathy and venous endotheliopathy promote the same inflammatory and hemostatic molecular pathogenesis. The clinical expression of the activated inflammatory pathway is similar to the variable cytokine effects with fever, malaise, arthralgia, and myalgia in both the arterial and venous vascular systems. However, the consequence on the genesis of the microthrombi strings from the activated microthrombotic pathway (i.e., VMTD (EA-VMTD)) is significantly different between the arterial and venous systems. Bacterial sepsis causing arterial endotheliopathy is more likely associated with TTP-like syndrome [5,6], characterized by the triad of thrombocytopenia, MAHA, and MODS. Conversely, during the COVID-19 pandemic, we learned that TTP-like syndrome has been extremely uncommon, supporting the prominent role of venous EA-VMTD. However, COVID-19 viral sepsis and its vaccine complication in spite of severe ARDS were typically associated with ITP/ITP-like syndrome characterized by “silent” isolated thrombocytopenia and occasional mild MAHA and MODS [13,14]. Consumptive thrombocytopenia resulting from microthrombogenesis in arterial endotheliopathy and venous endotheliopathy produces two different thrombocytopenic phenotypes: ITP-like syndrome, and TTP-like syndrome, due to their vascular characteristics (Table 4 and Figure 5). This difference provides us with fascinating insights into the role of the diversity of the vascular system.

Certainly, (1) anatomical (capillary/arteriolar vs. venule/sinusoid), (2) hemodynamical (high and low pressure/shear stress), (3) histological (different effects of ECs and SET), (4) functional (oxygen delivery vs. CO_2_ disposal), and (5) circulatory directional (efferent vs. afferent from and to the heart) differences exist between the arterial system and venous system. These anatomic and functional disparities in the two vascular systems and traveling of microthrombi in the efferent or afferent directions from or to the heart generate TTP-like syndrome or ITP-like syndrome in endotheliopathy. These characteristics produce “silent” venous microthrombosis in circulation, triggering ITP-like syndrome and pulmonary arterial capillary microthrombosis (i.e., ARDS) in venous endotheliopathy [13,14], but they induce capillary and arteriolar microthrombosis, triggering TTP-like syndrome with MAHA and MODS in arterial endotheliopathy [5].

### 3.3. TTP-like Syndrome and ITP-like Syndrome

Acquired immune-associated TTP and hereditary TTP, hemolytic-uremic syndrome (HUS), and TTP-like syndrome are considered thrombotic microangiopathies [44]. This conceptual term implies thrombosis-induced vascular damage resulting in thrombocytopenia, MAHA, and MODS. However, this concept has been revisited because HUS and TTP-like syndrome with well-established endothelial molecular pathogenesis are different from TTP that is characterized by severe ADAMTS13 deficiency [5]. In contrast to TTP, in TTP-like syndrome the endotheliopathy resulting from the activated complement system releases ULVWF/FVIII and activates the ULVWF path of hemostasis (Figure 1 and Figure 2), in which ULVWF multimers anchored to the endothelial membrane recruit platelets and generate microthrombi strings composed of platelet–ULVWF complexes, leading to EA-VMTD [5]. Thus, consumptive thrombocytopenia is the result of microthrombogenesis and MAHA and MODS are the consequence of microthrombosis (i.e., EA-VMTD) which partially occludes the arterial microvasculature of various organs. Certainly, the overexpression of the ULVWF/VWF antigen and increased factor VIII activity triggering thrombocytopenia are the result of endothelial damage, and they are the pathognomonic diagnostic markers for endotheliopathy.

In arterial edotheliopathy, if mild to moderate ADAMTS13 insufficiency is present in an individual due to a heterozygous mutation or the single nucleotide polymorphism of the ADAMTS13 gene, or due to the excess release of ULVWF from ECs over the ADAMTS13 capacity, TTP-like syndrome is produced with the triad of thrombocytopenia, MAHA, and MODS. To the contrary, in the same circumstances venous endotheliopathy ITP-like syndrome is promoted with isolated thrombocytopenia, as explained in Table 4. In the past, ITP-like syndrome and TTP/TTP-like syndrome have been considered as two completely different and unrelated diseases, but with the concept of endothelial molecular pathogenesis, their similarity is remarkable because they produce the same consumptive thrombocytopenia even though the hematologic phenotypes are dissimilar at different vascular milieux. The spectrum of consumptive thrombocytopenic purpura overlapping ITP/ITP-like syndrome and TTP/TTP-like syndrome in endotheliopathy is shown in Figure 5. The following are additional clinical data that support the hypothesis that both TTP-like syndrome and ITP-like syndrome are the result of the same endothelial molecular pathogenesis:TTP-like syndrome and ITP-like syndrome (e.g., most cases of ITP) are associated with thrombocytopenia, complement activation [45,46,47], and elevated VWF antigen/FVIII and ADAMTS13 insufficiency [10,48,49,50,51,52]. *Interpretation*: These changes are consistent with the activated complement system promoting endotheliopathy that leads to the exocytosis of ULVWF/FVIII, consumptive thrombocytopenia, and microthrombosis. However, microthrombosis occurs in the microvasculature in arterial endotheliopathy, as seen in TTP-like syndrome (i.e., aEA-VMTD), but it is commonly “silent” in venous circulation, as seen in ITP-like syndrome in venous endotheliopathy (i.e., vEA-VMTD);Despite thrombocytopenia, both are associated with the thrombophilic state [53,54,55,56]. *Interpretation*: TTP-like syndrome is already in the “microthrombotic state” within the microvasculature, and ITP-like syndrome is likely in “silent” microthrombotic state because the microthrombosis occurring in the venous system does not produce MAHA and MODS. However, acute venous thromboembolism (VTE), which is venous combined micro–macrothrombosis, can develop in ITP-like syndrome if additional venous vascular injury occurs, and especially in the intensive care unit (ICU) [14];Both have shown a beneficial response to therapeutic plasma exchange (TPE), intravenous immunoglobulins (IVIG), and rituximab [57,58,59,60,61,62,63,64]. *Interpretation*: The therapeutic benefit of these plasma therapies and anti-immune therapy suggests that both aEA-VMTD (TTP-like syndrome) and vEA-VMTD (ITP-like syndrome) share the same pathogenesis due to activated complement-associated endothewliopathy.

It is not uncommon that, in many human diseases, endotheliopathy may variably affect the arterial and venous systems, even in the same disease and in each patient. Therefore, endotheliopathy can produce combined TTP-like syndrome or ITP-like syndrome, which also could have been called TCIP [4]. For example, COVID-19 sepsis primarily causes venous endotheliopathy, producing ITP-like syndrome, but microthrombi strings in the venous circulation are mostly trapped in the pulmonary microvasculature, producing ARDS [7], which sometimes causes thrombocytopenia, mild MAHA, and rarely MODS in some patients.

Thrombocytopenia in clinically ill patients could be present with mild TTP-like syndrome, ITP-like syndrome, or combined TTP-like syndrome and ITP-like syndrome due to the variable endotheliopathy. It should be emphasized that the mechanism of “thrombocytopenia” in the majority of human diseases is not due to decreased production, increased immune destruction, or splenic sequestration, but is due to increased consumption/utilization following microthrombogenesis forming microthrombosis, which is promoted by endotheliopathy [4].

### 3.4. Endotheliopathic Inflammatory Syndromes

Endothelial inflammatory syndromes are characterized by the release of cytokines from damaged ECs that directly provokes inflammation and promotes the secretion of chemokines, which attract immune cells, such as monocytes and macrophages. A variety of terms have been used to define local and systemic inflammatory syndrome associated with endotheliopathy, as follows:Organ-designated endotheliitis, which is called multiorgan inflammatory syndrome (MOIS) [13] (e.g., myocarditis, pericarditis, encephalitis, pneumonitis, thyroiditis, hepatitis, nephritis, pancreatitis, adrenalitis, myositis, lymphadenitis, and others);Systemic inflammatory response syndrome (SIRS) [6];Cytokine storm/cytokine release syndrome [65];Multisystem inflammatory syndrome in child (MIS-C) [66];Multisystem inflammatory syndrome in adult (MIS-A) [67];Kawasaki disease [68];Polyarteritis nodosa [69];Acute necrotizing fasciitis [70];Thromboangiitis obliterans [71];Localized endothelial inflammatory syndromes (e.g., Sjogren’s syndrome, temporal arteritis, Crohn’s disease, Grave’s diseases, Hashimoto’s thyroiditis, Alzheimer’s disease, amyotrophic lateral sclerosis, myasthenia gravis, diabetic endotheliitis, rheumatoid endotheliitis, and others);Focal, local, regional, or disseminated endotheliitis/angiitis/vasculitis;Angiodysplasia/hemangiomatosis/telangiectasia/atrio-venous malformation.

The current interpretation is that these syndromes are the direct results of acute, subacute, or chronic endotheliitis, with or without extended vasculitis from the released inflammatory cytokines [65,72], such as interleukins, interferons, tumor necrosis factors, adhesive molecules, and chemokines, and the additional contribution of microthrombosis resulting in either arterial and/or venous endothelial damage [13].

### 3.5. Endotheliopathy-Associated Vascular Microthrombotic Disease

Endothelial microthrombotic syndromes are associated with the exocytosis of two of the most important coagulation factors: ULVWF and FVIII [11,73] from ECs, which recruit platelets to initiate partial hemostasis (microthrombogenesis) to form “microthrombi strings” that become anchored to the endothelial membrane of the intravascular vessel wall. Because both systemic or multiple local endothelial damages due to sepsis and other critical illnesses, as well as localized endothelial lesions, are limited to ECs, without damage to SET (Table 2), no hemorrhage occurs into the intravascular lumen. Instead, hemostasis is partially triggered by the activated lone ULVWF path [6], which produces disseminated or localized intravascular microthrombi strings within the microvasculature. This microvascular microthrombosis (i.e., EA-VMTD) orchestrates not only TTP-like syndrome and ITP-like syndrome, but also MODS and many other clinical microvascular microthrombotic syndromes, as follows.

#### 3.5.1. Examples of “Suspected” or Proven Endotheliopathy-Associated Microthrombotic Syndromes

##### Focal, Multifocal, Local, or Regional EA-VMTD [11,12]

Hereditary disease in early life
-Hereditary hemorrhagic telangiectasia/Osler–Weber–Rendu syndrome;-Capillary arterio-venous malformation syndrome;-Hereditary neurocutaneous hemangiomatosis;-Hereditary endotheliopathy, retinopathy, nephropathy and stroke (HERNS) syndrome;-Kasabach–Merritt syndrome with kaposiform hemangioendothelioma;-Fabry disease with endothelial dysfunction due to α-galactosidase A deficiency;Acquired disease in later life
-Susac syndrome with encephalopathy, retinopathy, and audiopathy;-Transient ischemic stroke (TIA) due to atrial-fibrillation-associated endotheliopathy [40,74];-Vascular dementia due to multifocal endotheliopathy associated with cognitive dysfunction;-Retinal microaneurysm in diabetes;-Degos disease associated with skin papules [75];-Henoch–Shoenlein purpura with subcutaneous hemorrhage and renal failure [76];-Heyde’s syndrome associated with aortic stenosis and intestinal angiodysplasia [77];-Alzheimer’s disease with chronic endotheliopathy-associated microtubule dysfunction [78,79];-Disseminated EA-VMTD [6,12];Critical-illness-associated endotheliopathy;Sepsis-associated endotheliopathy (e.g., pathogens, including bacteria, viruses, fungi, rickettsia, and parasites) [6];Chemical-/toxin-associated endotheliopathy (e.g., drug, toxin, vaccine, poison, and venom);Pregnancy-associated endotheliopathy (e.g., gestational thrombocytopenia; hemolysis, elevated liver enzymes, low platelets [HELLP] syndrome; preeclampsia; abruptio placenta; placenta previa; amniotic fluid embolism; dead fetus syndrome);Cancer-associated endotheliopathy (e.g., breasts, stomach, and lungs);Polytrauma-associated endotheliopathy (e.g., car accident);Surgery-associated endotheliopathy (e.g., open heart, abdomen, pelvic, and orthopedic) [80,81,82];Transplant-associated endotheliopathy (e.g., kidneys, heart, bone marrow, and stem cells) [83];Autoimmune disease-associated endotheliopathy (e.g., systemic lupus erythematosus (SLE) [84];Diabetic endotheliopathy due to plasma glycated CD59.

##### Organ/Multiorgan Dysfunction Syndromes

Encephalopathic stroke due to microthrombosis (e.g., diffuse encephalopathic stroke [40], diabetic ketoacidosis [5]);Reye’s syndrome with possible hepatic encephalopathy with MODS [85];ARDS due to pulmonary capillary endotheliopathy [7,14];Transfusion-related acute lung injury (TRALI) (possible consumptive thrombocytopenia-associated pulmonary artery endotheliopathy causing ARDS) [86];Subacute bacterial endocarditis (SBE) (consistent with inflammatory syndrome and microthrombosis due to endotheliopathy) [87];Veno-occlusive disease (VOD)/sinusoidal obstruction syndrome (SOS) consistent with hepatic venous endotheliopathy [88];Microvascular myocardial infarction (MVMI) due to diffuse microvascular microthrombosis [89];Fulminating hepatitis/acute liver failure due to diffuse microvascular microthrombosis, often with arterial endotheliopathy [90,91];Acute necrotizing pancreatitis, often with arterial endotheliopathy [92];Waterhouse–Friderichsen syndrome due to adrenal microvascular microthrombosis and hemorrhage [93];Acute renal failure (ARF)/hemolytic-uremic syndrome (HUS) [94];Goodpasture syndrome with pulmonary renal syndrome with ADAMTS13 insufficiency [95];Rhabdomyolysis with suspected arterial/venous endotheliopathy [96,97];Idiopathic pulmonary hypertension with pulmonary arterial endotheliopathy and microtubule dysfunction [98];Scleroderma with dermal endotheliopathy [99,100];Primary biliary cirrhosis with endothelial transformation;Felty syndrome with the overexpression of VWF (?);EA-VMTD with hepatic coagulopathy (old term: “acute DIC”) [8];Paroxysmal nocturnal hemoglobinuria;Combined organ syndromes with recognized terms (e.g., hepatorenal syndrome, hepatic encephalopathy, cardio-pulmonary syndrome, pulmonary encephalopathy, pulmonary-renal syndrome, and cardio-renal syndrome);MODS [6].

##### Systemic Syndromes

TTP-like syndrome (consistent with arterial EA-VMTD) [5];ITP-like syndrome/immune ITP (consistent with venous EA-VMTD) [13,14];Systemic lupus erythematosus (SLE) (consistent with inflammatory cutaneous endotheliopathy and MODS associated with EA-VMTD) [101];Antiphospholipid antibody (APLA) syndrome consistent with venous combined micro–macrothrombotic syndrome (e.g., endotheliopathy-associated VTE in pregnancy) [14,102];Kawasaki disease (consistent with inflammatory and lymphocutaneous syndrome in vaccine-induced venous endotheliopathy, and rarely with arterial endotheliopathy) [103];Malignant hypertension with hypertensive encephalopathy [104];Purpura fulminans (consistent with EA-VMTD with sepsis-associated endotheliopathy, and combined protein C and ADAMTS13 deficiency) [10,105];Graft-versus-host disease (consistent with post-allotransplant endotheliopathy) [106,107];Polyarteritis nodosa, acute necrotizing fasciitis, and thromboangiitis obliterans associated with combined inflammation and arterial endotheliopathy [69,70,71];Raynaud’s disease [108] caused by arterial endotheliopathy complicated by regional macrothrombosis from additional vascular injury [13].

##### Combined Micro–Macrothrombotic Syndromes [6,10,13,14]

Peripheral gangrene syndromes associated with arterial endotheliopathy and additional in-hospital arterial vascular injury (e.g., symmetrical peripheral gangrene (SPG), Fournier’s disease, Buerger’s disease, gas gangrene, diabetic gangrene, acute necrotizing fasciitis, Raynaud’s phenomenon, “coumadin”-associated gangrene syndrome, and envenomation syndrome);Venous circulatory congestion (VCC) syndromes (i.e., VTE, CVST) associated with venous endotheliopathy and additional in-hospital venous vascular injury (e.g., APLA syndrome, vaccine-induced thrombocytopenia and thrombosis syndrome) [13,14].

#### 3.5.2. Pathogenesis of Combined Micro–Macrothrombotic Syndromes [10,13,14]

Combined micro–macrothrombotic syndromes are complex phenotypes formed from underlying microthrombosis in endotheliopathy and additional fibrin clots resulting from another vascular injury due to vascular accesses, and especially in hospital and ICU settings. In clinical medicine, venous combined micro–macrothrombosis (e.g., VTE) is much more common than arterial combined micro–macrothrombosis (e.g., SPG) because the central venous catheter is more commonly utilized in the hospital and ICU. These syndromes are produced by the unifying process of the “microthrombi strings” of EA-VMTD and the “fibrin meshes” from vascular injury in the hospital/ICU that become “combined micro-macrothrombi” [6,10,13,14]. The complex micro–macrothrombosis can occur in the arterial system or venous system with two distinctly different clinical phenotypes. In the digits of the peripheral arterial system, the arterial microthrombi strings encounter fibrin meshes produced by additional traumatic arterial injury (e.g., indwelling arterial catheter) traveling efferently of the blood circulation. Both microthrombi and fibrin meshes produce numerous minute “combined binary micro-macrothrombi” by the unifying mechanism (Figure 2), and they occlude similar-sized peripheral small arterial vasculatures as a shower of the minute forms of combined micro–macrothrombi, leading to peripheral multiple digit gangrene [6,10]. The typical case is SPG. In “silent” microthrombi (ITP-like syndrome) in the venous system, the “microthrombi strings” encounter “fibrin meshes” formed at the additional venous injury site (e.g., indwelling central venous catheter) and form multiple, large, and extended venous “combined binary micro-macrothrombi” at the venous injury site due to slow venous circulation. These localized or regionalized combined venous micro–macrothrombi (i.e., VTE) travel afferently to the heart and lungs, causing pulmonary thromboembolism (PTE) [13,14], as seen in the COVID-19 pandemic. Therefore, in combined arterial or venous micro–macrothrombotic syndromes, inflammation can be a significant component due to the association with the underlying disseminated endotheliopathy [109].

#### 3.5.3. What Should We Have to Know about the Pathogenesis of EA-VMTD?

EA-VMTD is a hemostatic disease caused by the activation of ULVWF path of hemostasis;It is characterized by increased ULVWF/VWF/VWF antigen expression and increased FVIII activity due to the release from damaged ECs;It is associated with platelet consumption and the formation of microthrombi strings, and it may trigger microvascular hemolysis due to the increased shear-stress effect on the red cells, and especially in the arterial system (i.e., MAHA);It is often associated with the insufficiency of the ULVWF-cleaving enzyme ADAMTS13 secondary to its gene mutation or polymorphism, or excessive exocytosis of ULVWF over the cleaving capacity of ADAMTS13 in the endothelial damage;EA-VMTD orchestrates MODS, such as encephalopathy, ARDS, MVMI, acute necrotizing pancreatitis, rhabdomyolysis, hepatic dysfunction, acute renal failure, and sometimes multifocal microvascular microthrombosis with clinically significant phenotypes, such syndromes as retinal microaneurysm, TIA, Heyde’s syndrome, HERNS syndrome, and Susac syndrome.

### 3.6. Epiphenomenon

An epiphenomenon is an inconsequential manifestation of abnormal clinical or laboratory features that results from a benign pathogenetic process in association with human diseases. Thus, an endothelial epiphenomenon commonly occurs without serious clinical phenotypes, and the pathologic implication may be trivial and insignificant when compared with autoimmune diseases such as autoimmune TTP (i.e., anti-ADAMTS13 antibody), autoimmune VWD (i.e., anti-VWF antibody), and autoimmune hemolytic anemia (i.e., ant-Rh epitopes).

Prior to the COVID-19 pandemic, this author had concluded that ARDS was manifested by EA-VMTD in the pulmonary microvascular system [7], and now this pathogenetic mechanism has been confirmed as disseminated pulmonary microthrombosis that results from the molecular event of the exocytosis of ULVWF and FVIII and activated ULVWF path of hemostasis triggered by endotheliopathy [7,8]. Additionally, more complex forms of macrothrombosis in COVID-19 infection (i.e., VTE/PTE and SPG/digital gangrene) and vaccine-induced thrombocytopenic thrombosis (e.g., CVST) have been encountered as forms of combined micro–macrothrombosis in arterial and venous systems resulting from the unifying mechanism of ULVWF path and TF path of hemostasis [10,13,14]. However, the unresolved mysteries have been the mechanism and role of the immune response in COVID-19 infection and vaccination, including positive antinuclear antibodies (ANA), anti-DNA antibodies, anti-phospholipid (PL) antibodies, anti-platelet factor 4 (PF4) antibodies, and many other autoanibodies [13,14]. A very important question is; is this autoimmune phenomenon in the viral sepsis and vaccination responsible for thrombocytopenia and complex thrombosis syndromes, such as VTE and arterial gangrene?

#### 3.6.1. Positive Anti-PF4 Antibodies and Anti-PL Antibodies

The platelet-related auto-antibody appearance coinciding with thrombocytopenia and thrombotic disorder is not an uncommon event, but their cause–effect relationship has not been established in sepsis (e.g., COVID-19), postvaccination, and other critical illnesses. In the pre-pandemic and COVID-19 pandemic eras, the prevailing opinion has been that thrombocytopenia and thrombosis were related to each other in the pathogenesis, and perhaps through thrombogenesis, and subsequent platelet consumption and autoimmune platelet destruction. Thus, when anti-PF4 antibody was positive in the absence of heparin exposure, it was called heparin-induced thrombocytopenia (HIT)-like syndrome [110,111,112]. Moreover, when the anti-PL antibody was inexplicably positive, APLA syndrome was suspected [113,114]. This conceptual interpretation is wrong-footed because, for example, thrombocytopenia and thrombosis (e.g., VTE, CVST) are the results of two different pathogeneses, as explained earlier [13,14]. Anti-PF4 antibodies and anti-PL antibodies have been detected in some patients with “thrombosis” as well as “without” thrombosis. Indeed, their concurrence has not been consistently demonstrated, and the mechanism of the antibodies causing both thrombocytopenia and/or macrothrombosis has not been proven in the COVID-19 medical literature [115,116,117,118,119,120]. No doubt, in endotheliopathy, thrombocytopenia is not the result of platelet destruction caused by platelet-associated antibodies, but it is the consequence of consumption via microthrombogenesis due to the activated ULVWF path of hemostasis [6,11,12].

#### 3.6.2. Endothelial Epiphenomenon

Endotheliopathy provoked by the activated complement system releases various endothelial biomolecules and leads to the activation of platelets. Perhaps the altered biomolecules and platelet components following inflammation, microthrombogenesis and microthrombolysis may trigger the adaptive immune response and produce nonspecific and specific autoantibodies, such as antinuclear antibodies (ANA), anti-DNA, endothelial cell antibodies, anti-PL antibodies, anti-PF4 antibodies, anti-β islet cell antibodies, and others. It can be postulated that microthrombi strings composed of platelet–ULVWF complexes anchored to the microvascular EC membrane cause additional hypoxic endothelial and tissue damage, and also microthrombi strings could become self-degraded by the microthrombolysis of ADAMTS13 or plasmin [121]. The altered biomolecules, such as PF4 chemokines from degraded platelets [122], phospholipids from endothelial cells, and other proteins from the damaged endothelium, could be altered in their epitope sequence(s) by the mechanism of post-translational modification [123], and oxidative damage may lead to neo-autoantigen formation [124,125]. The neo-autoantigen of partially altered protein or molecules of the microthrombi strings of the platelet–ULVWF complexes could become “altered self” or “foreign-like” molecules [125,126] and promote antibody formation via the adaptive immune system.

Should these antibodies be nonspecific against “autoantigen PF4”, the antibody will be rested as an “epiphenomenon” [13,127,128,129], but if some of the antibodies are very specific and serious against the neo-antigen such as “PF4-heparin” complex, then the patient could develop “true” heparin-induced thrombocytopenia with “white clot syndrome” [130,131]. If this thesis is correct, the positive test for autoantibodies does not necessarily mean that they are the cause of thrombocytopenia and thrombosis, or autoimmune disorder. Nonspecific autoantibodies even with epitope specificity, including the antibodies against the molecules of PF4 or PL alone, are unlikely to provoke autoimmune disease. Indeed, the combined syndrome of thrombocytopenia (i.e., consumptive) due to endotheliopathy and thrombosis (i.e., VTE and SPG) in sepsis associated with additional vascular injury in ICU has been interpreted as venous or arterial combined micro–macrothrombosis [13,14], rather than HIT-like thrombosis syndrome and/or APLA syndrome.

Endotheliopathy-associated autoimmune diseases, such as SLE associated with positive anti-dsDNA, HIT-like thrombosis syndrome associated with anti-PF4 antibodies, APLA syndrome associated with positive anti-PL antibodies, or type 1 diabetes associated with positive anti-β islet cell antibodies, should be reexamined for their roles in the pathogenesis of autoantibody formation and clinical phenotypes, and particularly in light of the proven endothelial pathogenesis.

## 4. Practical Diagnostic Criteria and Diagnostic Perspective for Endotheliopathy

Recognizing endotheliopathy as the underlying pathogenetic mechanism of a clinical disease is a difficult task to prove because no unique clinical feature can be identified to make its specific phenotype diagnosis. Each clinical and pathological disease is so variable, even amongst the same clinical disorders. However, a guideline is summarized in Table 5 to suspect or confirm endotheliopathy as the underlying pathologic process based on its clinical, laboratory, and molecular findings.

Each clinical phenotype manifested by endotheliopathy is influenced by: (1) the primary pathology of inflammation and microthrombosis affecting the arterial or venous vascular system; (2) various organ and tissue localization by endothelial heterogeneity and/or the tropism of the external insult; (3) an additional secondary modifying pathology due to a genetic disorder, such as thrombophilia or von Willebrand disease, and an acquired disease, such as sepsis or polytrauma.

In the past, when we had not recognized the endothelial nature of the pathologic process in many diseases, imprecise terms such as polyarteritis nodosa, thromboangiitis obliterans, acrocyanosis, SLE, scleroderma, acute necrotizing fasciitis, Kawasaki disease, Raynaud’s phenomenon, Buerger’s disease, Fournier’s disease, SPG, limb gangrene, gas gangrene, peripheral vascular ischemia, and diabetic gangrene, and so on, were used. It is about time for the clinician to redefine and reclassify many human diseases based on the taxonomic diversity by identifying endotheliopathy as a major pathologic entity that occurs as a result of abnormal hemostasis in vivo, with the consideration of the interaction amongst the etiologic, genetic, pathogenetic, and environmental variables.

## 5. Conclusions

Despite the critical pathologic nature of endotheliopathy in so many human diseases, its molecular pathogenetic mechanisms, which lead to many clinical syndromes and phenotypes, have not been well defined in the medical community to date. This shortcoming has delayed the establishment of the diagnostic criteria, hampered the early recognition of the endotheliopathic nature of many diseases, and impeded the identification of the pathogenesis and effective therapeutic regimens. Now, the molecular pathogenesis of endotheliopathy is established based on two novel hemostatic theories. Certainly, the endothelium plays a very important role in the well-being of the patient encountered in the practice of clinical specialty and subspecialty, including in infectious disease, cardiovascular disease, neurological disorder, gastrointestinal disease, renal disease, endocrine disease, metabolic disease, cancer, trauma, and blood disease, as well as surgery, obstetrics, dermatology, and critical care medicine. In this review article, attempted is identifying the pathogenesis of endotheliopathy contributing to many of the poorly defined clinical diseases, disorders, and syndromes. Because endotheliopathy compromises the normal physiological homeostasis of the body and the hemostasis of the involved vascular system, a new creative approach is proposed to classify endotheliopathy-associated human diseases as a separate pathogenetic entity in our educational and medical care system. This will enhance the diagnostic skill of clinicians, facilitate scientists’ research endeavors, save patient lives by establishing the diagnosis of endotheliopathy early and promoting theory-based clinical therapeutic trials.

## Figures and Tables

**Figure 1 medicina-58-01311-f001:**
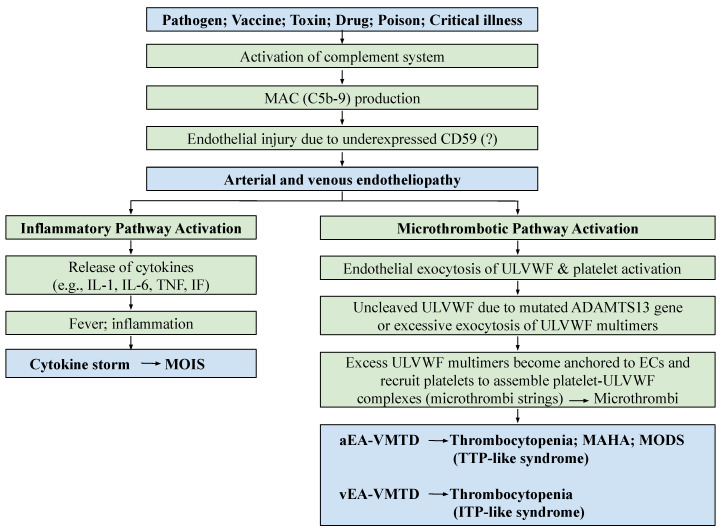
Pathogenesis of endotheliopathy based on “two-activation theory of the endothelium”. Endothelial molecular pathogenesis is initiated by the activated complement system following exposure to a pathogen, toxin, drug, poison, venom, vaccine, polytrauma, hyperglycemia, severe hypertension, and others. Endotheliopathy releases inflammatory cytokines and hemostatic factors, and activates two clinically important pathways: inflammatory and microthrombotic. Both arterial endotheliopathy and venous endotheliopathy provoke inflammation via the inflammatory pathway, leading to inflammatory syndrome, such as MOIS due to cytokines, but arterial endotheliopathy promotes microthrombosis via the microthrombotic pathway and produces aEA-VMTD due to the activation of the ULVWF path of hemostasis. Arterial endotheliopathy is characterized by the triad of thrombocytopenia, MAHA, and MODS, which is called “TTP-like syndrome”, but venous endotheliopathy is characterized by ITP/“ITP-like syndrome” due to silent microthrombi with consumptive thrombocytopenia, as explained in the text. The distinguishing features are caused by the different anatomies, physiological functions, and hemodynamic characteristics between arterial system and venous system. These are very important pathophysiological features in the understanding of the complexity in variable phenotypes amongst the endotheliopathic syndromes. Abbreviations: EA-VMTD: endotheliopathy-associated vascular microthrombotic disease; ITP: immune thrombocytopenic purpura; MAC: membrane attack complex; IF: interferon; IL: interleukin; MAHA: microangiopathic hemolytic anemia; MODS: multiorgan dysfunction syndrome; MOIS: multiorgan inflammatory syndrome; TNF: tumor necrosis factor; TTP: thrombotic thrombocytopenic purpura; aEA-VMTD: arterial EA-VMTD; vEA-VMTD: venous EA-VMTD; ULVWF: ultra-large von Willebrand factor.

**Figure 2 medicina-58-01311-f002:**
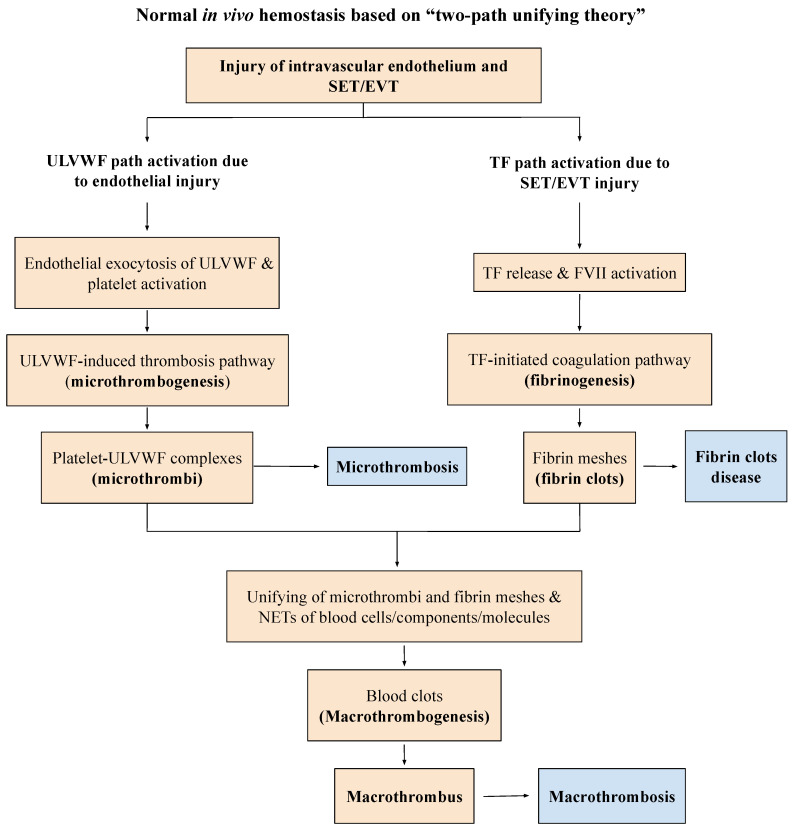
Normal in hemostasis in vivo based on “two-path unifying theory” (reproduced and modified with permission from the Chang JC. Thrombosis Journal. 2019;17:10). Following a vascular injury, the hemostatic system in vivo activates two independent sub-hemostatic paths: microthrombotic (ULVWF path) and fibrinogenetic (TF path). Both are initiated by the damage of ECs and SET/EVT due to external bodily injury and traumatic intravascular injury. In activated ULVWF path from EC damage, ULVWF multimers are released and recruit platelets, which produce microthrombi strings via microthrombogenesis, but in activated TF path from SET/EVT damage, released TF activates FVII. The TF–FVIIa complexes produce fibrin meshes/fibrin clots via fibrinogenesis of the extrinsic coagulation cascade. The final path of in vivo hemostasis is macrothrombogenesis, in which microthrombi strings and fibrin meshes become unified together with incorporation of NETs, including red blood cells, neutrophils, DNAs, and histones. This unifying event, “macrothrombogenesis”, promotes the hemostatic plug and wound healing in external bodily injury, and produces macrothrombosis in intravascular injury. Abbreviations: EA-VMTD: endotheliopathy-associated vascular microthrombotic disease: ECs: endothelial cells; EVT: extravascular tissue; NETs: neutrophil extracellular traps; SET: subendothelial tissue; TF: tissue factor; ULVWF: ultra-large von Willebrand factor multimers.

**Figure 3 medicina-58-01311-f003:**
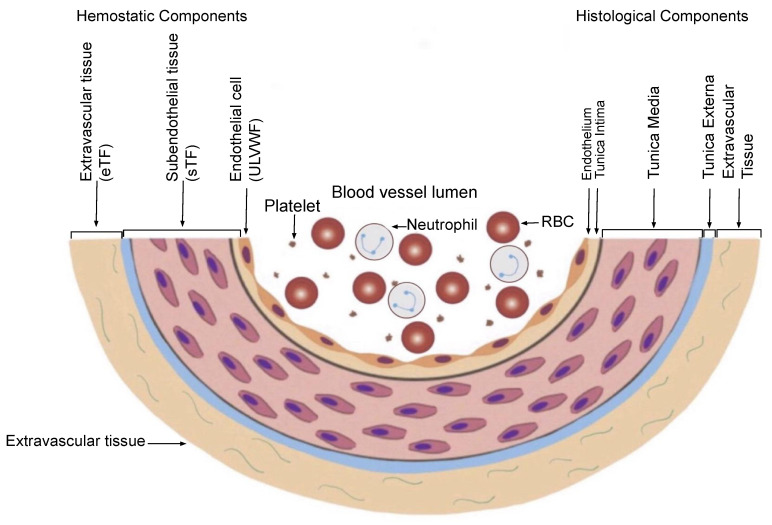
Schematic illustration of cross section of blood-vessel histology and hemostatic components (reproduced and modified with permission from Chang JC. *Clin Appl Thromb Hemost* 2019 January–December; 25:1076029619887437). The blood-vessel wall is the site of hemostasis (coagulation) in the external bodily injury at which blood clots (hemostatic plug) are produced and hemorrhage is stopped. It is also the site of hemostasis (thrombogenesis) in the intravascular injury in which intravascular blood clots (thrombus) are produced to cause thrombosis. Its histologic components are divided into the endothelium, tunica intima, tunica media, and tunica externa, and each component has its function that contributes to molecular hemostasis. As illustrated, EC damage triggers the exocytosis of ULVWF and SET damage, and it promotes the release of sTF from the tunica intima, tunica media, and tunica externa. EVT damage releases eTF from the outside of the blood-vessel wall. This depth of the blood-vessel-wall injury contributes to the genesis of different thrombotic disorders, such as microthrombosis, macrothrombosis, fibrin clots, thrombo-hemorrhagic clots, and various endotheliopathic syndromes. This concept based on the blood-vessel-wall model is especially important in the understanding of different phenotypes of stroke and heart attack. Abbreviations: EVT: extravascular tissue; eTF: extravascular tissue factor; SET: subendothelial tissue; sTF: subendothelial tissue factor; RBC: red blood cells; ULVWF: ultra-large von Willebrand factor.

**Figure 4 medicina-58-01311-f004:**
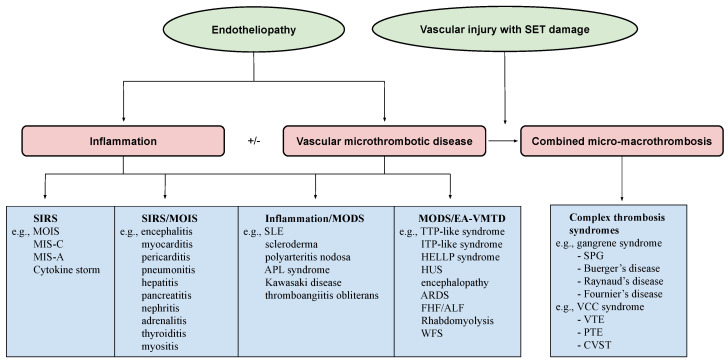
Projected spectrum of endotheliopathic syndromes in relation to endothelial pathogenesis and hemostasis. The illustrated figure shows that endotheliopathy is the underlying pathology of many human diseases, from inflammatory syndromes to vascular microthrombotic disease and combined micro–macrothrombosis, and it is manifested as many diseases on a spectrum. Because the roles of the endothelium and the contribution of the molecular mechanism of endotheliopathy as a hemostatic disease have not been clearly identified, the pathogenetic mechanisms of large parts of the endotheliopathic syndromes in this figure and the text have been considered to be mostly “unknown” or “idiopathic”. However, during the recent COVID-19 pandemic, the pathogenesis of ARDS was unmasked and confirmed to be the result of endotheliopathy leading to VMTD in COVID-19 sepsis. It was finally affirmed that endotheliopahy promotes a hemostatic disease via the lone activation of the ULVWF path of hemostasis as previously proposed [6,7]. Abbreviations: ALF: acute liver failure; APLA syndrome: antiphospholipid antibody syndrome; ARDS: acute respiratory distress syndrome; CVST: cerebral venous sinus thrombosis; EA-VMTD: endotheliopathy-associated vascular microthrombotic disease; HELLP: hemolysis, elevated liver enzymes, and low platelets; FHF: fulminant hepatic failure; HUS: hemolytic-uremic syndrome; MOIS: multiorgan inflammatory syndrome; MIS-C: multisystem inflammatory syndrome in child; MIS-A: multisystem inflammatory syndrome in adult; PTE: pulmonary thromboembolism; SIRS: severe inflammatory response syndrome; SLE: systemic lupus erythematosus; SPG: symmetrical peripheral gangrene; ULVWF: ultra-large von Willebrand factor; VTE: venous thromboembolism; WFS: Waterhouse–Friderichsen syndrome.

**Figure 5 medicina-58-01311-f005:**
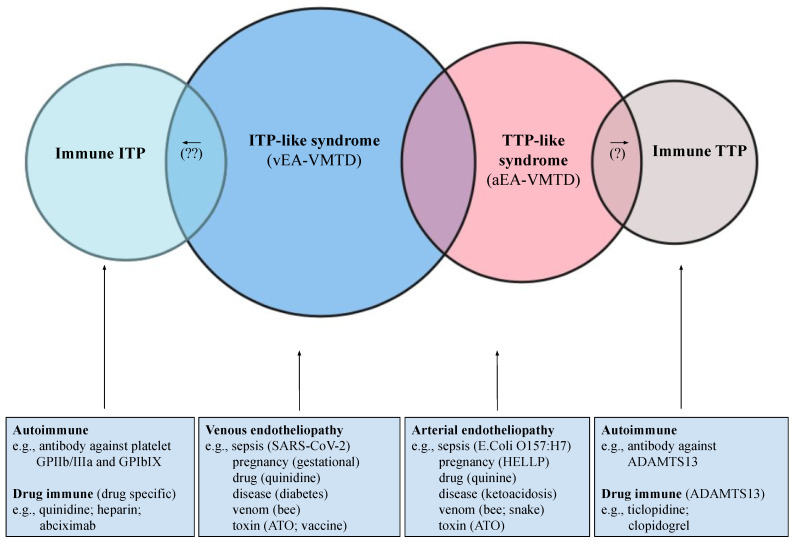
Projected distribution of acquired consumptive and immune-related thrombocytopenic syndromes (model based on endotheliopathy). Thrombocytopenia is often a mysterious origin in clinical medicine. It is very common in sepsis and critical illnesses, including diabetic complication, stroke, heart attack, autoimmune disease, hemostatic diseases, and of course, the side effects of many drugs. It has been typically interpreted as the condition due to decreased production from suppressed bone marrow, increased destruction due to the immune mechanism, or increased utilization, such as in diseases such as TTP-like syndrome. Still, the pathogeneses of a large number of thrombocytopenia could not be identified, and often itx has been called ITP, “idiopathic”, or “immune”. Because endotheliopathy has been found to be due to consumptive thrombocytopenia following microthrombogenesis in the lone ULVWF path of hemostasis, it has become clear that the largest number of thrombocytopenia is the result of consumption, which can be manifested differently in intersecting relationship between venous and arterial endotheliopathy, as well as immune-related mechanisms, and this clarifies the mystery of ITP/ITP-like syndrome and TTP/TTP-like syndrome. The figure is self-explanatory showing overlapping concepts. Abbreviations: ATO: arsenic trioxide; aEA-VMTD: arterial endotheliopathy-associated vascular microthrombotic disease; vEA-VMTD: venous EA-VMTD; HELLP: hemolysis, elevated liver enzymes, and low platelet syndrome; ITP: immune thrombocytopenic purpura; TTP: thrombotic thrombocytopenic purpura.

**Table 1 medicina-58-01311-t001:** Conceptual understanding of the terms: hemostasis, coagulation, and thrombosis.

	Hemostasis	Coagulation	Thrombosis
Term concept	Philosophical	Physiological	Structural
Implied meaning	Natural process in vivo	Artificial process in vitro Physiologic process at bleeding site	Pathological process in vivo
Involved site	Blood-vessel wall	Test tube Extravascular trauma to vessel wall	Intravascular lumen
Products	Hemostatic plug	Fibrin mesh/fibrin clot *	Microthrombi/macrothrombi
Critical role	Vascular-wall injury	Coagulation test Hemorrhage	Physiologic thrombogenesis
Components	Endothelium SET EVT Blood in circulation	TF/thromboplastin Coagulation proteins/serine proteases	ULVWF/FVIII from ECs Platelets^+^ from circulation Coagulation proteins/serine proteases TF from SET/EVT
Phenotypes	Determined by the level of vascular damage	Determined by participating coagulation factors	Determined by ULVWF, platelets, TF, hemostatic factors, and unifying mechanism
Inciting example	Endotheliopathy Vascular injury	Coagulation tests for PT and aPTT	VMTD/MODS due to microthrombosis EA-VMTD: TTP-like syndrome: “DIC” Arterial macrothrombosis VTE Combined micro–macrothrombosis

Abbreviations: “DIC”: false disseminated intravascular coagulation; DVT: deep vein thrombosis; ECs: endothelial cells; EVT: extravascular tissue; EA-VMTD: endotheliopathy-associated vascular microthrombotic disease; MODS: multiorgan dysfunction syndrome; PT: prothrombin time; aPTT: activated partial thromboplastin time; SET: subendothelial tissue; TF: tissue factor; TTP: thrombotic thrombocytopenic purpura; ULVWF: ultra-large von Willebrand factor multimers; VMTD: vascular microthrombotic disease; VTE: venous thromboembolism. * The only coagulation disorder in vivo is acute promyelocytic leukemia because only fibrin clots are formed without platelets.

**Table 2 medicina-58-01311-t002:** Three fundamentals in normal hemostasis.

(1) Hemostatic Principles		
(1) Hemostasis can be activated only by vascular injury; (2) Hemostasis must be activated through the ULVWF path and/or TF path; (3) Hemostasis is the same process in both hemorrhage and thrombosis; (4) Hemostasis is the same process in both arterial thrombosis and venous thrombosis; (5) The level of vascular damage (endothelium/SET/EVT) determines the different clinical phenotypes of hemorrhagic disease and thrombosis.		
(2) Major Participating Components		
Components	Origin	Mechanism
(1) ECs/SET/EVT	Blood-vessel wall/EVT	Protective barrier
(2) ULVWF	ECs	Endothelial exocytosis/anchoring and microthrombogenesis
(3) Platelets	Circulation	Adhesion to ULVWF strings/assembling and microthrombogenesis
(4) TF	SET and EVT	Release from tissue due to vascular injury/leading and fibrinogenesis
(5) Coagulation factors	Circulation	Activation of coagulation factors/forming and macrothrombogenesis
(3) Vascular Injury and Hemostatic Phenotypes		
Injury-Induced Damage	Involved Hemostatic Path	Level of Vascular Injury and Examples
(1) Endothelium	ULVWF	Level 1 damage—microthrombosis (e.g., TIA [focal]; Heyde’s syndrome [local); EA-VMTD [disseminated])
(2) Endothelium/SET	ULVWF + sTF	Level 2 damage—macrothrombosis (e.g., AIS; DVT; PTE; AA)
(3) Endothelium/SET/EVT	ULVWF + eTF	Level 3 damage—macrothrombosis with hemorrhage (e.g., THS; THMI)
(4) EVT alone	eTF	Level e damage—fibrin clot disease (e.g., AHS (e.g., SDH; EDH); ICH; organ/tissue hematoma)
Hemostatic Phenotypes	Causes	Genesis
(1) Hemorrhage	External bodily injury	Trauma-induced external bleeding (e.g., accident; assault; self-inflicted injury)
(2) Hematoma	Internal EVT injury	Obtuse trauma-induced bleeding (e.g., tissue and cavitary hematoma; hemarthrosis)
(3) Thrombosis	Intravascular injury	Intravascular injury (e.g., atherosclerosis; diabetes; indwelling venous catheter; surgery; vascular access)

Abbreviations: AA: aortic aneurysm; AHS: acute hemorrhagic stroke; AIS: acute ischemic stroke; DVT: deep venous thrombosis; ECs: endothelial cells; EDH: epidural hematoma; EVT: extravascular tissue; ICH: intracerebral hemorrhage; PTE: pulmonary thromboembolism; SDH: subdural hematoma; SET: subendothelial tissue; TF: tissue factor; eTF: extravascular TF; sTF: subendothelial TF; ULVWF: ultra-large von Willebrand factor; THS: thrombo-hemorrhagic stroke; THMI: thrombo-hemorrhagic myocardial infarction; TIA: transient ischemic attack.

**Table 3 medicina-58-01311-t003:** Three thrombogenetic paths and their clinical phenotypes of thrombosis according to the “two-path unifying theory of hemostasis”.

Thrombosis Mechanism	Microthrombogenesis	Fibrinogenesis	Macrothrombogenesis
Utilizing hemostatic path	ULVWF path	TF path	Combined ULVWF and TF path
Examples	Sepsis (bacterial, viral, fungal, rickettsial, parasitic) Polytrauma, surgery, transplant Toxin, drug, venom, vaccine Pregnancy Diseases (autoimmune, cancer, diabetes)	APL SDH EDH	DVT (distal) VTE (central DVT/PTE) Arterial thrombosis Peripheral gangrene
Thrombosis character	Microthrombi strings	Fibrin clots	Macrothrombus Combined micro–macrothrombi
Vascular lesion and phenotype example	Focal—retinal microaneurysm Local—hepatic VOD Multifocal—HERNS, Susac syndrome Disseminated—EA-VMTD/MODS	Disseminated—true DIC	Local—distal DVT, arterial thrombosis, AIS
Complex vascular/ hematologic phenotypes	Venous—venous microthrombi (ITP-like syndrome) Arterial—arterial microthrombi (TTP-like syndrome, MODS)	Venous/arterial—DIC (fibrin clot disease)	Combined micro–macrothrombosis Venous—VTE, PTE, CVST Arterial—SPG, limb gangrene

Abbreviations: AIS: acute ischemic stroke; APL: acute promyelocytic leukemia; CVST: cerebral venous sinus thrombosis; DIC: disseminated intravascular coagulation; DVT: deep venous thrombosis; EDH: epidural hematoma; MODS: multiorgan dysfunction syndrome; PTE: pulmonary thromboembolism; ITP: immune thrombocytopenic purpura; PTE, pulmonary thromboembolism; SDH: subdural hematoma; SPG: symmetrical peripheral gangrene; TF: tissue factor; TTP: thrombotic thrombocytopenic purpura; ULVWF: ultra-large von Willebrand factor; VOD, veno-occlusive disease; VTE: venous thromboembolism.

**Table 4 medicina-58-01311-t004:** Clinical phenotypes and mechanisms of endotheliopathy in arterial and venous systems per the “two-activation theory of the endothelium”.

Clinical Phenotype	Arterial Endotheliopathy	Venous Endotheliopathy
Underlying pathology	aEA-VMTD	vEA-VMTD
Physiological/hemodynamic difference	Efferent circulation from the heart (oxygen delivery) High-pressure flow High shear stress Capillary and arteriolar microvascular events	Afferent circulation into the heart (CO_2_ disposal) Low-pressure flow Low shear stress Venous and pulmonary microvascular events
Primary cause		
Vascular injury (ECs)	Sepsis-induced arterial microvascular endotheliopathy	Sepsis-induced venous endotheliopathy Vaccine-induced venous endotheliopathy
Vascular pathology site	Disseminated aEA-VMTD at the microvasculature	Transient or “silent” vEA-VMTD at the venous system
Activated hemostatic path	ULVWF path	ULVWF path
Thrombosis component	Microthrombi strings in the microvasculature	Microthrombi strings in the venous system
Microthrombotic event	Disseminated VMTD	Silent microthrombosis with efficient (?) microthrombolysis
Clinical phenotypes	TTP-like syndrome - consumptive thrombocytopenia - MAHA - MODS/MOIS	ITP-like syndrome - consumptive thrombocytopenia - atypical MAHA - ARDS/MOIS

Abbreviations: aEA-VMTD: arterial endotheliopathy-associated vascular microthrombotic disease; vEA-VMTD: venous-EA-VMTD; ECs: endothelial cells; ITP: immune thrombocytopenic purpura; MAHA: microangiopathic hemolytic anemia; MODS: multiorgan dysfunction syndrome; MOIS: multiorgan inflammatory syndrome; TTP: thrombotic thrombocytopenic purpura; ULVWF: ultra-large von Willebrand factor.

**Table 5 medicina-58-01311-t005:** Endotheliopathy: based on clinical, laboratory, and molecular characteristics.

**Clinical Features** Common in critically ill patients related to certain conditions (e.g., sepsis, polytrauma, surgery, transplant, drug, toxin, poison, venom, vaccine, pregnancy, autoimmune immunity, cancer, hypertension, diabetes, and various vasculopathies); Mucocutaneous manifestations (rashes, purpura, papules, telangiectasia, angioma/angiomatosis, erythema, eczema, hemorrhage); One or more organ dysfunction syndrome (e.g., MODS); Significant inflammation and sometimes MOIS, cytokine storm, MIS-C, or MIS-A; Endothelial epiphenomenon (e.g., positive anti-PF4 antibodies, anti-PL antibodies, ANA, anti-DNA); Microthrombosis (i.e., EA-VMTD), or arterial or venous combined micro–macrothrombosis (e.g., SPG, VTE); Trauma or ICU admission (e.g., in-hospital vascular injury).
**Laboratory and Molecular Features** Consumptive thrombocytopenia with TTP, ITP, or TCIP; Overexpression of ULVWF/VWF/VWF antigen; Increased FVIII activity; ADAMTS13 insufficiency due to its gene mutation or excessive exocytosis of ULVWF; ITP-like syndrome or TTP-like syndrome.
**Diagnostic Blood Tests** Platelet count; ULVWF/VWF/VWF antigen determination; FVIII activity; ADAMTS13 activity; Markers of endothelial epiphenomenon; Activated complement system (e.g., downregulated C5b-9); Cytokine increase (e.g., cytokines such as ILs, TNFs, IFNs and chemokines).

Abbreviations: EA-VMTD: endotheliopathy-associated vascular microthrombotic disease; ICU: intensive care unit; IL, interleukin; IFN, interferon, ITP: immune thrombocytopenic purpura; MIS-A: multisystem inflammatory syndrome in adult; MIS-C: multisystem inflammatory syndrome in children; MODS: multiorgan dysfunction syndrome; MOIS: multiorgan inflammatory syndrome; PF4: platelet factor 4; PL: phospholipids; SPG: symmetrical peripheral gangrene; TCIP: thrombocytopenia in critically ill patients; TNF, tumor necrosis factor; TTP: thrombotic thrombocytopenic purpura; ULVWF: ultra-large von Willebrand factor; VWF: von Willebrand factor.

## Data Availability

Data sharing is not applicable to this article as no datasets were generated or analyzed in the current study.

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
