# Peer review of "Molecular Pathogenesis of Endotheliopathy and Endotheliopathic Syndromes, Leading to Inflammation and Microthrombosis, and Various Hemostatic Clinical Phenotypes Based on “Two-Activation Theory of the Endothelium” and “Two-Path Unifying Theory” of Hemostasis"

_medicina, 2022, doi:10.3390/medicina58091311_

Round 1

Reviewer 1 Report

The manuscript titled Molecular pathogenesis of endotheliopathy and endotheliopathic syndromes, leading to inflammation and microthrombosis, and various hemostatic clinical phenotypes based on “two-activation theory of the endothelium” and “two-path unifying theory” of hemostasis presents the endotheliocentric view at vascular inflammation and microthrombosis.

The review has been prepared with good diligence and leads from the basics of endothelial function to its significance in different diseases. The work is solid, well outlined, and organized. Although the idea of such a review seems to be very important and the growing number of papers in this field indicates the need for summaries, the realization require some improvements. I suggest some modifications below:

-I recommend adding a list of abbreviations. It makes reading easier, especially in biological works where the number of abbreviations is significant.

- in the list of co-authors is given “Jae C. Chang and M.D.”. What does the abbreviation M.D. stand for?

- in such an extensive review I miss two aspects related to endothelial dysfunction: 1./ changes in endothelial stiffness upon inflammation/diseases (PMID: 29162916; PMID: 30550804), and 2./ formation of endothelial lipid droplets in response to inflammatory factors (PMID: 32084444).

Author Response

Author’s Response to Reviewer 1

1)   I have appreciated your kindly review of a long manuscript of mine containing the complexity of issues in “endotheliopathy and endotheliopathic syndromes”.

2)   You are correct that the manuscript has used many abbreviations for biomolecular terms and clinical ones. As you have recommended for the readership, I have added a list of abbreviations at the end of the main body of manuscript.

3)   I am the sole author and also corresponding author, which should be “Jae C. Chang”.

4)   I agree with your points that, in addition to vascular characteristics of the venous system vs. arterial system and the microvasculature vs. macrovasculature, vascular homeostasis affecting endothelial, vascular wall stiffness/elasticity upon inflammation and diseases also play an important role to the change of endothelial function and could contribute to endotheliopathic syndromes and clinical phenotypes. Certainly, we should learn more about this dynamic process. Since I am not familiar with vascular mechanical studies beyond the logical understanding of circulatory physiology of the vascular system and vascular tree, I have not included this specific issue in my manuscript. The vascular mechanics and functional understanding besides biological and molecular changes (e.g., TTP-like syndrome in arterial endotheliopathy and ITP-like syndrome in venous endotheliopathy) should be interesting research ideas affecting vascular system by interested scientists, especially in aging process.

5)   I am sincerely thankful to your favorable comments on my article, and also helping me to widen the scope of my vision on the potential of endothelial dysfunction that can be influenced by physical mechanics, lipid droplets and glycocalyx. I will be vigilant on these subjects in my future works.

Reviewer 2 Report

Topic was extensively covered.  Well written review article. 

However requires more original studies to strengthen the theories of endotheliopathy. 

Author Response

Author’s response to Reviewer 2

1)   Thank you for reviewing my long manuscript. I have sincerely appreciated the dedication of your time to journal Medicina in this review process. Your effort in review process is an important contributing factor for the progress of medical science, which also encourages diligence of many clinicians and scientists.

2)   The concept of endotheliopathy is unequivocally established now. It is true that, based on the skeletal structure of my article, clinical medicine would need further studies on the endothelial pathogenetic nature for many unresolved human diseases to strengthen the theories of endotheliopathy and hemostasis in vivo. Since its molecular hemostatic mechanism is well-established with endothelial markers and clinical phenotypes, we also should be able to venture into molecular vasculopathogenesis, atherosclerogenesis and angiogenesis as well as thrombogenesis, especially for the role of hemostasis in intravascular injury associated with autoimmune disease. Yes, it is only the beginning of new chapter on the endotheliopathy.

Reviewer 3 Report

The author proposes a narrative review regarding the molecular pathogenesis of endotheliopathy proposing the “two-activation theory of the endothelium” and the “two-path unifying theory” of hemostasis. The aim is to identify the mechanisms of microthrombosis and macrothrombosis with particular focus on disseminated vascular micro-thrombotic disease (VMTD), a term coined by the author  , also considering thrombotic complications of COVID-19.

Lines 105-109: the disease entities, which are considered as imprecise, are indeed possible complications of COVID19 in different organs.

    1. line 122: complement activation: as also expressed on line 146, complement is only one possible trigger of endotheliopathy. In fact endothelial cells are involved in the regulation of vascular permeability, leukocyte adhesion/ extravasation, vascular tone, and hemostasis with a strict interplay between these function in case of endothelial dysfunction which should be considered by the author. The endothelial layer is covered by a dense glycocalyx— a gel-like layer of proteoglycans and extracellular matrix components, which is involved in transendothelial transport,e.g., of lipoproteins, and which can be lost or reduced in inflammation.

Lines 178-180: the author refers to his own papers [5,6,8,20] without referring to specific experimental data.

Lines 197-200, lines 263,: this statement only refers to author’s own paper and it is debatable to support the hypothesis that endothelial cell do not express tissue factor (TF) when activated (as also shown in figure 1 and 4). Although TF is constitutively expressed in extravascular tissues, such as fibroblasts and smooth muscle cells [Bode MF, Mackman N. Protective and pathological roles of tissue factor in the heart. Hamostaseologie. 2014;35.] activated endothelial cells and adhered leukocytes may express active TF in response to vascular injury or inflammatory stimuli. Activated endothelial cells can express TF, which has an important role in the pathogenesis of thrombosis [Parry GC, Mackman N. Transcriptional regulation of tissue factor expression in human endothelial cells. Arterioscler Thromb Vasc Biol. 1995;15:61221.Borissoff JI, Spronk HM, ten Cate H. The hemostatic system as a modulator of atherosclerosis. N Engl J Med. 2011;364:174660.].TF may exist as an inactive (encrypted) form that becomes activated (decrypted) upon vascular injury [Chen VM, Hogg PJ. Encryption and decryption of tissue factor. J Thromb Haemost. 2013;11 Suppl 1:27784.] and this may account for the presence of TF protein with little to no procoagulant activity. As a result, endothelial cells can express TF which is a involved in the deposition of fibrin and thus to fibrinogenesis, and not only lead to in ULVWF/FVIII platelets “ microthrombi”. Therefore endothelial cells seem to have a role in both fibrinogenesis and ULVWF microthombosis, and they role cannot be separated as shown in figure 1 and 4.

Author Response

Author’s response to Reviewer 3

1)   I am very grateful to you for your review of my manuscript and constructive critique, and am happy to learn your special interest in the mechanism of hemostasis in vivo.

2)   Line 105-109: “an insult causing endothelial damage would disrupt normal anatomy and physiologic function. The endothelium may lead diversified paths of the pathogenesis triggering the dysfunction of the cell, tissue, organ and multisystem”.

Response: Traditionally, the concept of sepsis has been considered to be the disease of pathogen damaging “endothelium” of the host, however, the treatment of sepsis based on this restricted theory has failed in numerous clinical trials in medical community many decades until this very day. This failure was due to the wrong concept that bacteremia and viremia are the cause of sepsis. In truth, sepsis is the result of “toxin of the pathogen” leading to endotheliopathy promoting inflammation and microthrombogenesis.  I have clearly explained the molecular pathogenesis of sepsis in my original paper on “Sepsis and Septic Shock” (Ref. #6: PMID: 31160889) and COVID-19 Sepsis paper (Ref. #10: PMID: 34103921). Pathogen-induced infection has three phases in its pathogenetic process: 1) entry phase of pathogen into the tissue, 2) bacteremic/viremic phase into the blood, 3) septic phase of pathogen toxin (e.g., S-protein and others) triggering endotheliopathy via activated complement. The complement mechanism on the endothelium was first discovered by Hattori (Ref #1). I had wrestled with the intrigue of sepsis phenomenon and postulated “two-activation theory of the endothelium”. This endothelial hemostatic mechanism has explained two important endothelial phenomena: 1) inflammation and 2) microthrombogenesis, and molecular changes occurring in endotheliopathy. These clinical features (e.g., inflammation, microthrombosis, increased ULVWF/VWF antigen, increased FVIII, and insufficient ADAMNTS13) proven retrospectively in previously published articles and laboratory studies during COVID-19 pandemic. Still, the most of ID specialists do not understand the difference between bacteremia and sepsis, perhaps because of incomplete comprehension of hemostatic nature of the endotheliopathy.

Even bacteria are eliminated by antibiotics and viruses are no longer present after initial infection, sepsis continues to progress due to toxin-induced endotheliopathy, which eventually may lead to the demise of the patient due to TTP-like syndrome, hepatic coagulopathy and MODS. In endotheliopathy, the dysfunction of the cells, tissue, organ and multisystem occurs as a result of “hypoxia” due to microthrombosis (i.e., EA-VMTD).

3)   Line 122 and 146: “complement activation”. Complement is only one trigger? 

Response: Yes, complement is the main trigger. The review was published by Kerr et al. (Ref # 3: PMID: 21855165).  The “complement activation” is the mechanism causing endothelial damage most likely via endothelial membrane channel (pores) formation, which results in “molecular changes” (i.e., release of cytokines, ULVWF/FVIII, thrombomodulin, adhesive molecules, and others). This molecular event has been confirmed in endotheliopathy.

Increased vascular permeability is not the cause of endotheliopathy, but is the result of endothelial damage caused by microthrombosis. Leukocyte adhesion is not the primary event, but is likely secondary one caused by cytokines, Vascular tone is probably due to neural homeostasis in response to endothelial injury. Thus, the main event is endotheliopathy leading to molecular changes that orchestrate clinical phenotypes.

Past several years, I have been mystified by “glycocalyx layer” covering the intravascular membrane of the endothelium and “ULVWF strings” anchored at the endothelial membrane on their relationship (PMID: 31984614). Their function and morphology are similar on the hemostasis, and location and in the length of slimy layer and strings. Both are related to Golgi structure and are glycoproteins. Could they be the similar or same property due to differently expressed definition based on structural identity vs. molecular identity by biochemist and biologist?  I am neither the biochemist nor laboratory scientist. But I wonder ULVWF multimer strings may be a part of glycocalyx or vice versa. The study of the character between two biological molecules with immunochemical study, electron microscopy and functional evaluation would be a great interest in hemostasis.

In clinical medicine, clinicians have thought DIC and TTP-like syndrome are considered to be two different diseases until recent past, but now we know better both are the same disease (i.e., EA-VMTD).

4)   Lines 178-180: “the author refers to his own papers [5,6,8,20]” on hemostatic mechanisms.

Response: It is true that I have used my articles to cite in References. But I have had no choice because two theories are needed to explain in my next paper. I have postulated and confirmed two hemostatic theories. One is “two-path unifying theory” of hemostasis, and the other is “two-activation theory of the endothelium”. The endothelial exocytosis in endotheliopathy was published by Dr. Hattori et al (Ref. # 1). in his laboratory study, and later I have independently postulated “two-activation theory” by a logical deduction while researching on hemostasis. The complement role on the endothelial damage was published by “Kerr et al. (Ref.  #3), which I have credited to Dr. Kerr and her authors” in the previous publications. The two theories I have constructed have been proven time and again in my entire publications (e.g., sepsis, DIC, Stroke, ARDS, DVT/VTE, VHF, TTP-like syndrome) without exception, I am confident now that I can utilize in these two theories in search of pathogenesis in many poorly-defined human diseases as I am applying to my current manuscript on “endotheliopathy and endothiopathic syndroms”, which you can see in the title.

I wish I can cite others’ objective papers as well if someone else has published the similar theories in the literature that can support my manuscript. But none is available. That is only reason why I have cited my own articles to help Medicina readership. To tell the truth, I am happy to use my own proven theories to persuade scientists and also help clinicians. Further, I still can solve the pathogenetic mechanisms of many mysterious hemostatic diseases (e.g., PNH, pulmonary hypertension, diabetic ketoacidosis, atherosclerosis, etc.) along with the “Principles of hemostasis” published in ARDS paper (Ref. #7: PMID: 31775524)

5)   The debate on the source of TF from the endothelium

Response: First, “the endothelial cell” is not “endothelial tissue” and it has not been unequivocally proven to contain any TF. However, some scientists insisted TF had to be present in ECs to justify TF-FVIIa activated coagulation cascade to produce thrombosis when the concept of microthrombogenesis was not known. Other scientists still focus on the endothelium in vascular injury, but do not consider different characters between traumatic vascular injury causing ECs +SET+EVT damage and septic endothelial injury causing only ECs damage. The former releases ULVWF and TF, but the latter only releases ULVWF. They may equate vascular injury with endothelial injury, but logical concept of traumatic vascular injury causes ECs+SET+EVT damage. The concept that TF released into blood from little endothelial damage without intravascular bleeding must activate FVII path is wrong one. The endothelial injury alone produces only microthrombosis as discussed in detail in Ref. # 8 (PMID: 33061857) because macrothrombosis never occurs without fibrin clots following fibrinogenesis. Even extensive microthrombosis (e.g., EA-VMTD) with TTP-like syndrome does not cause fibrinogenesis. If TF is present in the blood with microthrombosis, the patient would develop extensive microthrombosis and macrothrombosis simultaneously everywhere in the vascular trees, causing immediate demise of the patients. Please understand that a bleeding from external bodily injury due to penetrating trauma would damage to the “skin + EVT+ SET + ECs” of the involved blood vessel. Since EVT and SET before penetration to ECs are the rich source of TF, TF would be positive in the blood sample if tested after a vascular access. When we take a blood sample with a needle from a person/animal to measure TF in plasma and blood cells (e.g., monocytes, neutrophils, macrophages), the full length of skin-EVT-SET-ECs gets damaged by the needle. So, minute amounts of TF would contaminate in the blood and blood cells. Little vascular injury depending upon vascular access technique creates “positive’ TF in the blood sample. I have not read of any experiment showing TF-related molecules on “purely” isolated ECs without penetrating SET-EVT-skin.

Second, as far as what I know of encryption and decryption theory, this theory has never been postulated with structurally defined caveolae with the storage mechanism of TF as well as the mechanism of transformation between encrypted and decrypted forms. Indeed, the research results on TF present in ECs are contradictory and theory is very controversial. If this theory is correct, how extensive microthrombosis composed of platelet-ULVWF complexes (e.g., ARDS) occur without evidence of coagulopathy participated by TF as seen in COVID-19. Also, if this theory is correct, why every patient with TTP-like syndrome does not develop massive microthrombosis-macrothrombosis-bleeding disorder in the entire vascular trees leading to life-threatening thrombo-hemorrhagic disorder?

After all, why does microthrombosis occur in endotheliopathy such as sepsis, polytrauma and diabetes, but DVT always occur as macrothrombosis in full length vascular wall injury (e.g., vascular access, surgery, device)? I have no doubt that the endothelium does not and should not contain TF because it is inconsistent with life in disseminated endotheliopathy. Thus, microthrombosis is the result of endotheliopathy and macrothrombosis is the result of the entire vascular wall damage from trauma. Nature has provided human with “selectively” functioning hemostatic mechanisms.

6)   My thanks to you. I have very much appreciated and understood your challenging questions and debate. I still have to learn more from critiques and certainly you have stimulated my thinking process. My best wishes to you.

Jae C. Chang, M.D.